# How to Evaluate the Level of Green Development Based on Entropy Weight TOPSIS: Evidence from China

**DOI:** 10.3390/ijerph20031707

**Published:** 2023-01-17

**Authors:** Xiang-Fei Ma, Ru Zhang, Yi-Fan Ruan

**Affiliations:** 1School of Marxism, China University of Geosciences, Wuhan 430074, China; 2School of Public Administration, China University of Geosciences, Wuhan 430074, China

**Keywords:** new development concepts, green development, entropy weight TOPSIS, regional differences, China

## Abstract

Evaluating the level of green development is of great significance to better implement the concept of green development. By constructing an evaluation index system for green development, this paper comprehensively uses the entropy weight Technique for Order Preference by Similarity to an Ideal Solution (TOPSIS) method and coefficient of variation method to evaluate the green development level of 30 provinces in China from 2010 to 2019 and analyzes the regional differences of green development in China. The research findings are as follows: First, the level of green development in China is low but shows a slow rise trend, from 2010 to 2019; China’s green development level rises from 0.274 to 0.317, an increase of 15.7%. Secondly, regional differences of green development in China are obvious, with the level ranking from high to low as eastern, western, and central regions. Third, regional differences in China’s green development first widen and then narrow, with the variation coefficient of green development in 30 provinces and eastern, central, and western regions of China showing an inverted U-shaped trend of first increasing and then decreasing. Fourth, the regional difference of green development in eastern China is largest, followed by western China, and the smallest is central China. Finally, based on research findings, relevant policy recommendations are put forward.

## 1. Introduction

Over the past 40 years of reform and opening up, China’s economy has developed rapidly. In 2010, China’s GDP (gross domestic product) surpassed Japan for the first time, ranked second place in the world [1]. Although China has achieved great success in economic development over the past 40 years, China’s economy has been facing increasing downward pressure since 2012. In 2012, China’s GDP growth rate was 7.8%, and before 2012, China’s GDP growth rate rarely fell below 8% [2]. The downward pressure of China’s economy mainly comes from the increasingly serious aging population in China, increasingly strengthened to control environmental pollution and the sluggish consumer demand. In particular, the lack of technological innovation and the deterioration of ecological environment exposed in the process of China’s economic development have attracted people’s attention. The increasingly frequent environmental pollution problems have endangered the health of the people seriously [3]. In order to cope with the increasing economic downward pressure and environmental crisis occurring on economic development, in 2015, the Chinese government announced that China’s economic development had entered a new normal, and in order to adapt to the new normal of economic development, Chinese government proposed new development concepts. New development concepts included green, openness, coordination, innovation, and sharing, and these development concepts aimed to address various problems in China’s development [4]. The Fifth Plenary Session of the 18th CPC Central Committee put forward the new development concept of innovative development, coordinated development, green development, open development, and shared development. The new development concept is put forward in response to the new problems emerging in China’s economic development. The new development concept has aroused great concern after it was put forward in 2015. China’s central government urged local governments to speed up the implementation of the new development concept. Among the new development concepts, the concept of green development was put forward to deal with the imbalance between humans and environment. It is estimated that the average economic loss of 190 cities caused by environment pollution in China in 2014–2016 was 0.3% of the GDP [5]. Environmental crises have damaged the public’s trust in the Chinese government severely. Therefore, how to implement the concept of green development, improve the level of green development, and reverse the worsening trend of environmental pollution in China has become a topic of general concern to the Chinese government and people. Based on the new development concept proposed by the Chinese government, this paper aims to design the index system of green development, then use the entropy weight TOPSIS (technique for order preference by similarity to an ideal solution) method to evaluate the level of green development in China from 2010 to 2019. This paper is beneficial to accurately understanding the actual situation of green development in China and is also conducive to China’s more scientific implementation of the concept of green development. Compared with the existing literature, the marginal contributions of this paper are as follows: first, based on the new development concept, the technology innovation factors are incorporated into the green development to enrich the research about evaluating the green development level. In addition, the comparison of the level of green development in two different periods before and after China clearly put forward the concept of green development is also helpful to understand the current situation of green development in China. Second, the coefficient of variation (CV Index) is used to analyze the variation in regional differences in green development, which is beneficial to the research of coordinated regional development.

The rest of the paper is as follows. The second part is a literature review about green development. The third part is the research design of this paper, including an introduction of the entropy weight TOPSIS method and the coefficient of variation, green development index system, and data sources. The fourth part is the research results and discussion. The fifth part is the research conclusion, which presents a summary of this paper.

## 2. Literature Review

The research on green development can be traced back to the 1970s, where people realized the disadvantages of the economic development model at the cost of resource consumption. People then began to think about how to coordinate the relationship between economic development and ecological environment, and the concept of green sustainable development was born [6]. In recent years, the concepts related to “green” have emerged, including green economy, green development, green, and low-carbon [7,8,9]. The concept of green development aroused wide attention in the “Green New Deal” initiative proposed by the United Nations Environment Program in 2008 [10]. Green development not only emphasizes economic development but also ecology protection, resources conservation, and improving public welfare, which has become an inevitable requirement for high-quality economic development [11]. Chinese government put forward a new development concept for the first time in 2015. Green development is an important part of the new development concept. It is closely related to society, economy, and environment and was attached importance by the Chinese government [12,13]. Green development has not only been mentioned many times in the relevant policies of the Chinese government but also implemented in practice by the Chinese government, and the Chinese government made unremitting efforts to improve the level of green development. In recent years, academic circles have conducted a lot of research on green development. Reviewing existing literature, it is found that the research about green development mainly focuses on the construction and evaluation of green development indicators, the realization path of green development, and influencing factors.

On the one hand, in terms of the research on assessing the level of green development, Li et al. constructed green development indicators from the dimension of living environment, economic growth, pollutant treatment and utilization, innovation potential, and ecological efficiency. Then, they used the S-type cloud model to measure the green development level of the Beijing-Tianjin-Hebei region and found that the pollutant treatment and utilization level of the Beijing-Tianjin-Hebei region were high and that the development level of the other four indicators was low [14]. Wang et al. introduced health factors into the study of green development for the first time and constructed green development indicators from economy, the environment, and health; then, the weight cloud model was adopted to analyze the green development level of China’s listed mineral resources companies. It was found that the green development levels of most listed companies of mineral resources were low [15]. Han et al. selected 17 indicators based on the three dimensions of economy, resources, and health; used the entropy weight method and spatiotemporal analysis method to evaluate the green development level of the Association of Southeast Asian Nations (ASEAN) member states; found a higher economic development level of ASEAN member states and higher green development level [16]. Adetama et al. added low-carbon to the economic, social, and environmental dimensions to build a green development indicator system; then, the entropy TOPSIS method was used to assess the level of green development in 34 provinces of Indonesia [17]. In addition, many scholars have paid attention to the studies of rural green development. For example, Tao and Xiang constructed indicators from four dimensions of green economy, investment, utilization, and security to calculate the level of rural green development in Hunan Province by using the entropy weight method [18]. Considering regional differences in agricultural green development, Hou and Wang constructed an index system from four dimensions of agricultural development, resource protection, environmental friendliness, and industrial extension and integration to calculate the agricultural green development level of the three northeastern provinces by using the entropy weight method and grey correlation method and found that the agricultural green development level of the three provinces in Northeast China fluctuated and rose [19].

On the other hand, in terms of the research on the implementation path and influencing factors of green development, scholars put forward suggestions to improve the level of green development mainly from two aspects of economic transformation and environmental protection. First, from the economic perspective, green development is influenced by human resources, urbanization, green technology progress, green finance, and other factors. Based on the data of China’s A-share listed companies from 2008 to 2017, He et al. adopted the panel data regression model to analyze the impact of the academic background of top managers on corporate green innovation and found that the higher the education level of top managers, the stronger the green innovation ability of enterprises and the better the green development level of enterprises [20]. Zhang and Zhu believed there is a close relationship between urbanization efficiency and green growth and the improvement of urbanization efficiency can promote green economic growth [21]. Kosepglu et al. found that green innovation is the best solution for green sustainable development. A 1% increase in green innovation technology can reduce the ecological footprint by 0.129% [22]. In terms of financial development, Afzal et al. conducted a regression analysis on the panel data of 40 European countries from 1990 to 2019 and found that outbound investment was significantly positively correlated with environmental deterioration, and financial development can improve environmental quality then promote green development [23]. Secondly, from the perspective of environmental protection, factors affecting green development include infrastructure construction, environmental protection investment, environmental regulation, and vegetation coverage. Pauleit et al. pointed out that promoting the construction of green infrastructure can make a great contribution to the sustainable development of cities [24]. Feng et al. found that improving the vegetation coverage rate is conducive to improving the ecological environment and promoting the development of economy and environmental protection [25]. In addition, green investment is becoming an important driving force for green and sustainable development in China [26]. Zhang et al. adopted the nonlinear autoregressive distributed lag (NARDL) model to analyze the relationship between green investment, natural resource rent, and ecological footprint in China from 2000 to 2018, and found that green capital has a significant positive correlation with natural resource rent and ecological footprint [27]. Some scholars further analyzed the impact of environmental regulations on green development and found that environmental regulations are conducive to the progress of green technology, but with the deepening of environmental regulations, the impact of environmental regulations on green technology innovation in different regions has gradually expanded [28].

Through the literature review, it can be found that the current academic research on green development has made a series of rich achievements, but there are some deficiencies. First, the construction of a green development evaluation index system mainly focuses on economy, society, and environment, and lacks the consideration of the technical dimension. In recent years, with the development of artificial intelligence, 5G, new energy development, and other technologies, the impact of technology on the economy, society, and ecological environment is growing. Therefore, it is necessary to include technological innovation in the evaluation indicators of green development. Secondly, the existing literature focuses on the analysis of the differences in green development between regions, while little literature considers the variation in the differences in green development within regions. Therefore, this paper constructs green development indicators from four dimensions of green economy, green ecology, green society, and green technology based on panel data of 30 provinces in China from 2010 to 2019; then, the entropy weight TOPSIS model and coefficient of variation (CV Index) are used to measure China’s green development level and regional differences. Finally, based on the research findings, this paper puts forward policy suggestions to better promote the improvement of green development level.

## 3. Methodology

### 3.1. Assessment of Green Development Level

The TOPSIS method is a finite scheme multi-attribute decision-making method [29]. Entropy weight TOPSIS is a hybrid model based on the entropy method and the TOPSIS method, and its main feature is to determine the weight of each evaluation indicator based on the entropy weight method and then use the technique of approaching the ideal solution to determine the ranking of evaluation objects [30]. The basic idea of the entropy weight TOPSIS method is to determine the ideal solution, that is, all attribute values have reached the optimal (or inferior) value in the alternative scheme, and then by measuring the relative distance between each evaluation object and the optimal solution and the worst solution, the evaluation object is optimal if it is closest to the optimal solution and farthest away from the worst solution; otherwise, it is not optimal. The entropy weight TOPSIS method can make full use of the information of original data, has no special requirements on the sample size, and has the advantages of less information loss and flexible operation [31]. The calculation process of entropy weight TOPSIS is shown in Figure 1.

The specific calculation process is as follows.

Assuming that there are *m* evaluation objects and *n* evaluation indicators for each evaluated object, a judgment matrix is constructed (see Formula (1)).
(1)$$X=(x_{ij})m\times n\;i=1,2,\ldots ,\;m;\;j=1,2,\ldots ,n$$

First, all indicator data need to be standardized to eliminate the impact of different properties and units of indicators. The specific data standardization process is shown in Formulas (2) and (3).
(2)$$\mathrm{Positive}\;\mathrm{indicators}:\;x^{’}=\frac{x_{ij}-min(x_{ij})}{max(x_{ij})-min(x_{ij})}$$
(3)$$\mathrm{Negative}\;\mathrm{indicator}:\;x^{’}=\frac{max(x_{ij})-x_{ij}}{max(x_{ij})-min(x_{ij})}$$

In Formulas (2) and (3), *x_ij_* represents the index data; *max(x_ij_)* and *min(x_ij_)* are the maximum and minimum values, respectively, in the index data.

Second, the weight of each indicator is measured.

In the process of calculating the weight of indicators, the weight of the *i*-th evaluation object of the *j*-th indicator should be calculated first (see Formula (4)).
(4)$$P_{ij}=\frac{x_{ij}}{\sum_{i=1}^{m}x_{ij}}$$

Then, the entropy *e_j_* of the index is calculated (see Formula (5)).
(5)$$e_{j}=\left[-\frac{1}{\ln\left(m\right)}\right]\sum_{i=1}^{m}P_{ij}\ln\left(P_{ij}\right)$$

Finally, the weight *w_j_* of the index is calculated (see Formula (6)).
(6)$$w_{j}=(1-e_{j})/\sum_{j=1}^{n}\left(1-e_{j}\right)$$

Thirdly, the Euclidean distance and closeness are calculated.

The maximum and minimum values of each index in the decision matrix are written as the optimal solution vector $X^{+}$ and the worst solution vector $X^{-}$, respectively (see Formula (7)).
(7)$$X^{+}=(max\;x_{i1},max\;x_{i2},\ldots \ldots max\;x_{ij})\;X^{-}=(max\;x_{i1},max\;x_{i2},\ldots \ldots max\;x_{ij})$$

The entropy weight Euclidean distance is used to calculate the distance between each object and the worst solution and the optimal solution $d_{i}^{-}$, $d_{i}^{+}$ (see Formulas (8) and (9)).
(8)$$d_{i}^{-}=\sqrt{\sum_{i=1}^{j}w_{j}{\left(x_{ij}-x_{j}^{-}\right)}^{2}};i=1,2\ldots j;\;0\leq d_{i}^{-}\leq 1$$
(9)$$d_{i}^{+}=\sqrt{\sum_{i=1}^{j}w_{j}{\left(x_{ij}-x_{j}^{+}\right)}^{2}};i=1,2\ldots j;\;0\leq d_{i}^{+}\leq 1$$

Finally, the closeness between each ratio and the state of the relative object is calculated based on the two distance values of $d_{i}^{+}$ and $d_{i}^{-}$ (see Formula (10)).
(10)$$\partial_{i}=\frac{d_{i}^{-}}{d_{i}^{+}+d_{i}^{-}};\;i=1,2\ldots \ldots m,\;0\leq \partial_{i}\leq 1$$

In Formula (10), the larger the value of $\partial_{i}$, the better the status of the comparison object and the closer the processing index of *i* is to the ideal value. If each index of the comparison object is in the optimal state, then $\partial_{i}$ = 1. If each indicator of the comparison object is in the worst state, then $\partial_{i}$ = 0.

### 3.2. Assessment of the Regional Differences of Green Development

The coefficient of variation (CV index), also known as the standardization rate, is usually used to reflect the degree of dispersion of sample data. The CV index is the ratio between the standard deviation of the original data and the mean of the original data [32]. In this paper, the coefficient of variation is used to calculate the regional differences of green development in eastern, central, and western China and nationwide. In this paper, the larger the coefficient of variation is, the greater the difference of regional green development level is, and the more unbalanced regional development is; conversely, the smaller the difference of regional green development level is, the more balanced regional development is. The specific calculation formula is as follows.
(11)$$CV=\sqrt{\frac{1}{m}\sum_{k=1}^{m}\left(C-\overset{-}{C}\right)}/(\overset{-}{C})$$

In Formula (11), *m* represents the number of samples, and *C* represents the green development level of each province calculated by the entropy weight TOPSIS method. $\overset{-}{C}$ represents the average of green development levels in the eastern region, central region, and western region of China and nationwide.

### 3.3. Design of Green Development Index

To evaluate the level of green development, corresponding indicators should be selected. At present, there are many studies assessing the level of green development. Long et al. built an evaluation index system to meet the requirements of green development of coal-resource-based cities from the four dimensions of green economic development, green social development, green resource development, and green environmental development [33]. Han et al. constructed the index system of green development from economy and society, resources and utilization, and environment and health [16]. With the development of technology, the impact of technology on economy, society, and environment is increasingly prominent. Therefore, refer to the research of Wu et al. [34]. In addition, considering the availability of data, this paper constructs the index system of green development level from green economy, green ecology, green society, and green technology (see Table 1).

First, in terms of green economy, economic development involves the input and output of resources; therefore, referring to the research of Li et al. and Zhang et al. [14,35], the proportion of science and technology expenditure in fiscal expenditure and the proportion of energy conservation and protection expenditure in fiscal expenditure are selected for green input; in terms of green output, per capital GDP, the proportion of added value of tertiary industry in GDP, and per capital disposable income of residents are selected. Secondly, in terms of green ecology, resource utilization is the premise of sustainable development of a natural system and ecological protection is the basis of sustainable development of a natural system. Therefore, the index selection of green ecology is from resource utilization and ecological protection. Referring to the study of Xue et al. [36], the indicators of resource utilization include per capital water resources, per capital forest stock, and forest coverage rate. Indicators for ecological protection include the proportion of newly increased soil erosion control area in the area under jurisdiction, the reduction rate of SO_2_ emissions, and the reduction rate of chemical oxygen demand emissions. Thirdly, the index of green society is selected from two dimensions of green life and green consumption. Referring to the study of Zhang et al. [37], the index of green life includes the green coverage of built-up areas, the total passenger traffic of urban public trams, and the harmless disposal rate of household waste; indicators of green consumption include the growth rate of urban per capital natural gas consumption and the decline rate of per capital water consumption. Finally, in terms of green technology, the progress of technology often requires resource input and is reflected in technological output. Therefore, green technology indicators are selected from the two dimensions of technology input and technology output. The indicators of technology input are selected as the proportion of R&D expenditure in GDP of industrial enterprises above the designated size, and the full-time equivalent of R&D personnel in industrial enterprises above the designated size and the indicator of technology output are selected as the number of patent grants. After the above analysis, a green development index system has been established (see Table 1).

### 3.4. Data Sources

All data used in this study are from the official information released by the National Bureau of Statistics of China, which mainly includes the China Statistical Yearbook, China Statistical Yearbook of Science and Technology, and China Statistical Yearbook of Energy from 2011 to 2020. In addition, the green development level of only 30 provinces and cities in China was evaluated because some data for Taiwan, Hong Kong, Macao, and Tibet were missed.

## 4. Research Results

### 4.1. Assessment of Green Development Level

Based on the green development indicators in Table 1, the entropy weight TOPSIS method was used to evaluate the green development level of 30 provinces in China from 2010 to 2019. At the same time, in order to more clearly show the status of green development in China, 30 provinces in China were divided into eastern regions, central regions, and western regions. The results are shown in Table 2.

It can be seen from the results of Table 2 that the overall level of green development in China was low but showed a slow upward trend. From 2010 to 2019, the overall level of China’s green development increased from 0.274 in 2010 to 0.317 in 2019, an increase of 15.7%. Although China’s green development level in 2019 was higher than that in 2010, it can be seen from Table 2 that China’s green development level was highest in 2014, reaching 0.400. From 2010 to 2014, China’s green development showed a rising trend, then reached the maximum in 2014, and began to decline in 2015. In 2015, China put forward a new development concept, including the concept of green development. From 2015 to 2019, the level of green development in China showed a trend of fluctuation decline. China’s green development by 2019 was 0.317. Although it grew 15.7% in 2010, its annual growth rate was low. This shows that China has a long way to go in pursuing green development and needs to further implement the concept of green development. Specifically, the Chinese government needs to increase investment in science and technology and capital, to promote the transformation of China’s development from factor-driven to innovation-driven.

It can be seen from Table 2 that there are obvious regional differences in China’s green development and the green development level is ranked as eastern region, western region, and central region from highest to lowest. It can be seen from Table 2 that the green development level in the eastern region exceeds the national average, while the green development level in the central and western regions is lower than the national average, which indicates that China’s green development is uneven among different regions. The root cause may be that the eastern region has the dual advantages of policy and economy; the higher the level of economic development, the more reasonable the industrial structure; thus, the eastern region has made great progress in green development [38]. The green development level of central and western regions is lower than the national average, and the green development level of western regions is higher than that of central regions. The reason is that, on the one hand, the promotion of the western development strategy provides policy support for upgrading the industrial structure in the western region, which not only improves the level of economic development, but also reduces environmental pollution, and then promotes the green development of the western region to a certain extent [39]. On the other hand, the economic development level of the central region is lower than that in the eastern region, and the resource richness is lower than that of the western region, so the green development level is lowest. In Table 2, comparing the green development level of the provinces in the eastern region, central region, and western region, the green development level of Guangdong province in the eastern region was highest, rising from 0.461 in 2010 to 0.508 in 2019, while the green development level of Hebei Province in the eastern region was lowest, rising from 0.200 in 2010 to 0.259 in 2019. In western China, Qinghai and Inner Mongolia have a high level of green development; in 2019, the green development levels of Qinghai and Inner Mongolia were 0.395 and 0.357, respectively. Xinjiang and Gansu have a low level of green development; in 2019, the green development levels of Xinjiang and Gansu were 0.247 and 0.188, respectively. This indicates that the levels of green development in the eastern, central, and western regions differ greatly not only among each other, but also within the eastern, central, and western regions. Therefore, the coefficient of variation was used to analyze the green development level in eastern, central, and western China and in China as a whole.

### 4.2. Comparison of the Different Stages of Green Development

In Section 4.1 of this paper, the entropy weight TOPSIS method is adopted to measure the green development level of 30 provinces and cities in China, and it is found that from 2010 to 2019, the green development level of China has fluctuated and increased. It can be seen from Table 2 that 2014 was the turning point of China’s green development. In 2014, the green development level of most provinces in China reached its maximum, and then began to decline from 2015. In 2015, the Chinese government made it clear that it would implement the new development concept, and the concept of green development belongs to the new development concept. Therefore, in this part, China’s green development from 2010 to 2019 was divided into two stages: 2010 to 2014 and 2015 to 2019, and the two stages were compared and analyzed (see Table 3).

From Table 3, it can be more intuitively found that from 2010 to 2019, China’s green development showed obvious stage characteristics; from 2010 to 2014 is the first stage and 2015 to 2019 is the second stage. It can be seen from Table 3 that although China’s green development level in 2019 was higher than that in 2010, the highest green development level in China from 2010 to 2019 was 2014. From 2010 to 2014, the green development level of the eastern region, central region, and western region and the whole country increased from 0.326, 0.228, 0.254, and 0.274, respectively, to 0.463, 0.362, 0.331, and 0.388 in 2014, increases of 42%, 59%, 30%, and 42%, respectively. However, since the Chinese government proposed the implementation of the concept of green development in 2015, China’s green development began to decline. Although China’s green development level began to increase from 2016, it did not recover to the level of 2013 and 2014. Why did the Chinese government put forward the concept of green development in 2015, but the level of green development in China declined in 2015? The reason may be that in order to implement the concept of green development, the Chinese government strengthened the supervision of ecological environment; especially during this period, severe haze appeared in the north of China. As a result, the Chinese government faced pressure of ecological and environmental deterioration. Therefore, governments at all levels of China have strengthened environmental supervision, which has brought pressure on China’s economic growth. However, facing pressure of economic downturn after 2015, the Chinese government adjusted measures in time, prompting China’s green development to enter the stage of recovery. In general, more research is needed on the root causes behind this phase of green development in China in the future.

### 4.3. Analysis of Regional Differences of China’s Green Development

Section 4.1 calculates the green level of 30 provinces and cities in China from 2010 to 2019 by using the entropy weight TOPSIS method (see Table 2). This study divided 30 provinces and cities in China into eastern, central, and western regions, and analyzed the green development level of these three regions. In order to clearly map the spatial and temporal pattern of green development level in different regions of China, ArcGIS software was used in this paper to draw the spatial distribution map of green development level in 30 provinces of China in 2010, 2014, 2015, and 2019 with equidistant values. The darker the color, the higher the level of green development (see Figure 2).

From Figure 2, the change in green development level of each province and city in China can be found more directly. First, by comparing Figure 2a–d, it can be found that the color of Figure 2b is darkest, which means that the green development level of different regions in China was highest in 2014, which is consistent with the research results in Table 2. In addition, it can also be seen from Figure 2 that the color of central China is obviously lighter than that of eastern and western China, that is, the green development level of central China is lower than those of eastern and western China. It can also be seen from Figure 2 that the green development level of China’s eastern coastal areas is significantly higher than those of other provinces in China. In addition, there is a large difference between the green development level of provinces in central and western regions. Therefore, the coefficient of variation was adopted in this paper to analyze the intra-regional differences of green development levels in different regions of China.

This part uses the coefficient of variation to analyze the differences in green development in eastern China, central China, western China, and the whole country. Based on the results of green development in different regions of China measured in Table 2, Formula (11) was used to calculate the coefficient of variation of regional differences in China’s green development (see Figure 3 for results).

From Figure 3, it can be seen that the variation coefficient of green development in eastern, central, and western China during 2010–2019 showed an inverted U-shaped trend of first increasing and then decreasing. It can be seen from Figure 3 that after 2014, the variation coefficient of China’s green development level gradually stabilized. In other words, the regional difference of green development in China showed a trend of first expanding and then decreasing. This shows that the Chinese government has achieved remarkable results in narrowing the regional development gap. It can be seen from Figure 3 that the variation coefficient of China’s green development from high to low is the eastern region, western region, and central region; in addition, the coefficient of variation of green development in eastern China is close to that of the whole country. This shows that, on the one hand, the eastern region first put forward the development of a low-carbon circular economy, which forced technological innovation to continuously boost green development; together with the large amount of capital, talents, and scientific and technological foundation accumulated in the early stage, the eastern region laid the material foundation for green development [40]. On the other hand, the green development level of the Yangtze River Delta region in the eastern region is much higher than those of other provinces in the eastern region. The Yangtze River Delta region takes the lead in entering the stage of high-quality economic development; in the process of rapid economic development, it can respond to the green policy and encourage enterprises to fulfill their environmental protection responsibilities [41]. The eastern region of China should pay attention to the unbalanced green development within the region in the future while implementing the concept of green development. From Figure 3, the variation coefficient of green development in western China fluctuates greatly and shows a change track of first rising and then declining. The coefficient of variation in western China reached the highest value of 0.456 in 2013, and since 2013, the coefficient of variation has decreased significantly, to 0.198 in 2019, with a decrease of 56.6%. This shows that the differences in green development among different provinces in western China gradually narrowed. From Figure 3, it can be seen the central region has the lowest coefficient of variation in green development, and the coefficient of variation in green development from 2010 to 2019 was lower than the national average, which is consistent with the research results of Zhang et al., (2022) [42].

## 5. Conclusions

Based on the panel data of 30 provinces and municipalities in China from 2010 to 2019, this paper constructs the green development index system and comprehensively uses the entropy weight TOPSIS method to measure the green development level of China; then, it further uses the coefficient of variation to explore the regional differences of green development in the eastern, central, and western regions. The research findings are as follows: first, from 2010 to 2019, the level of green development in China was low but showed a slow rising trend; the overall level of green development in China rose from 0.274 in 2010 to 0.317 in 2019, an increase of 15.7%. Second, there were significant differences in the level of green development in different regions of China; the level of green development in the eastern region and western region was higher than that of the central region. Thirdly, the variation coefficients of green development in China and eastern, central, and western regions showed an inverted U-shaped trend of first increasing and then decreasing, which shows that the regional differences of China’s green development first widened and then narrowed. In addition, the highest coefficient of variation for green development in China is the eastern region, followed by the western region, and the lowest is the central region. The coefficient of variation of green development level of the eastern region is close to the national level, while the coefficient of variation of the central region is lower than the level of national average. Compared with the existing studies on the assessment of the level of green development, the largest contribution of this paper lies in the comparison of the level of green development in two different stages before and after China put forward the concept of green development. In addition, some studies only used the entropy weight method, the TOPSIS method, or the grey relational degree method, but did not combine the entropy weight method and TOPSIS method. Therefore, the adoption of the entropy weight TOPSIS method to assess the green development level of different provinces and cities in China is also different from other studies [18,19]. Based on the research findings, in order to promote China’s green development, the following suggestions are put forward.

First, Chinese government needs to adhere to the new development concept and pay more attention to green development. The results of this paper show that although the level of green development in China has increased, the overall level still low, only reaching 0.317 by 2019, indicating that China still has a long way to go to implement the concept of green development. Therefore, the Chinese government needs to further strengthen support to implementing the concept of green development. In particular, on the one hand, the government of China should strengthen the financial support for science and technology enterprises to help their development; on the other hand, the supervision of environmental pollution enterprises should be strengthened to accelerate the transformation and upgrading of environmental pollution enterprises and eliminate backward production capacity. Secondly, it can be found from the research results of this paper that from 2010 to 2019, China’s green development level and coefficient of variation from high to low were in the eastern region, the western region, and the central region. There is a large difference among different regions, and the green development gap in eastern, western, and central regions is also large. Therefore, China needs to pay attention to the green development gap between the central and western regions and the eastern regions, and narrow the intra-regional green development gap in eastern, central, and western regions. In particular, the government in the eastern region needs to strengthen the financial transfer payment to the central and western regions, and strengthen the talent support from the eastern region to the central and western regions.

## Figures and Tables

**Figure 1 ijerph-20-01707-f001:**
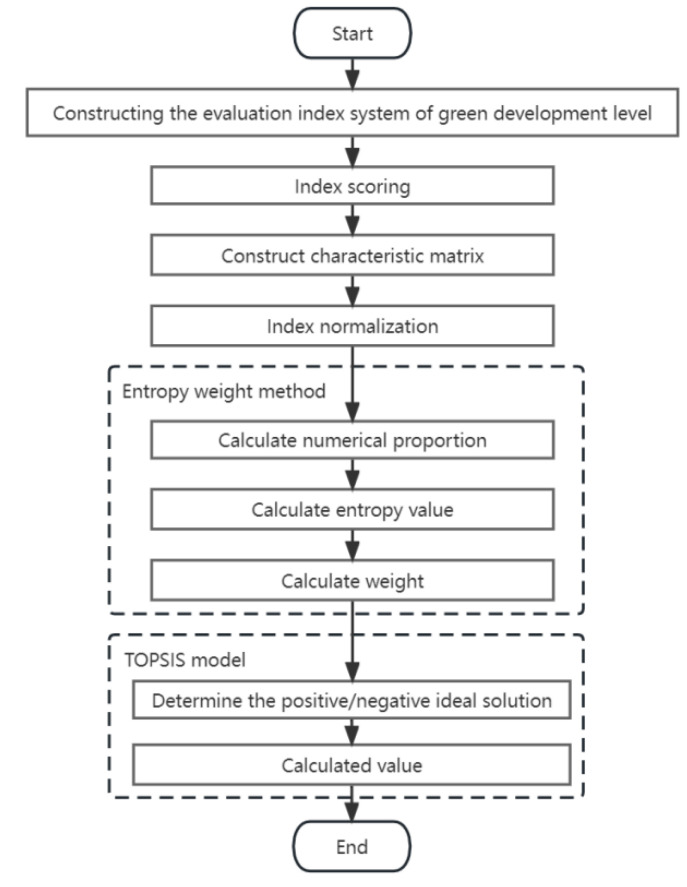
The calculation process of the entropy weight TOPSIS method.

**Figure 2 ijerph-20-01707-f002:**
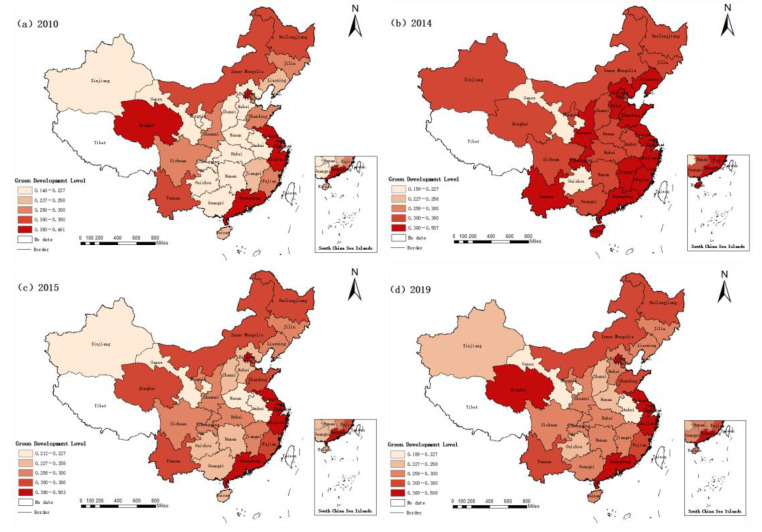
Spatial and temporal patterns of China’s green development in 2010, 2014, 2015 and 2019.

**Figure 3 ijerph-20-01707-f003:**
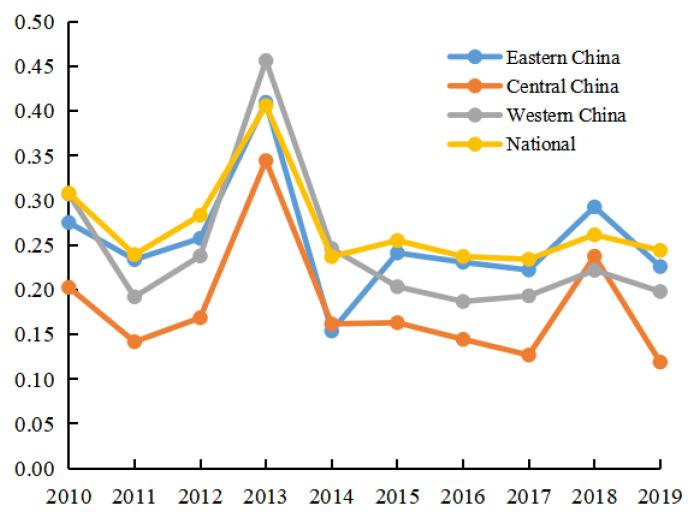
Variation coefficient of China’s green development level from 2010 to 2019.

**Table 1 ijerph-20-01707-t001:** The index system of green development.

Target Index	First Level Index	Second Level Index	Third Level Index (Unit)	Weight
Level of green development	Green economy	Green input	Expenditure on science and technology as a proportion of government expenditure (%)	0.066
			Expenditure on energy conservation and protection as a proportion of government expenditure (%)	0.042
		Green production	Per capital GDP (Yuan)	0.056
			Proportion of tertiary industry added value in GDP (%)	0.040
			Per capital disposable income of residents (yuan)	0.057
	Green ecology	Utilization of resources	Per capital water resources (cubic meters/person)	0.143
			Per capital forest stock (cubic meters/person)	0.101
			Forest coverage rate (%)	0.035
		Conservation of ecology	Proportion of newly added area under control of soil erosion in the area under jurisdiction (%)	0.063
	Green society	Green live	Green coverage rate of built-up areas (%)	0.042
			Total passenger volume of urban public trams (10,000 passengers)	0.038
			Harmless treatment rate of household garbage (%)	0.028
		Green consumption	Urban per capital natural gas consumption growth rate (%)	0.021
			Per capital water consumption decline rate (%)	0.017
	Green technology	Investment in technology	Proportion of R&D expenditure in GDP of industrial enterprises above designated size (%)	0.079
			Full-time equivalent of R&D personnel of industrial enterprises above designated size (person)	0.023
		Technical output	Number of patents granted (pieces)	0.110

**Table 2 ijerph-20-01707-t002:** Green development level in China from 2010 to 2019.

Region	Province/Year	2010	2011	2012	2013	2014	2015	2016	2017	2018	2019
Eastern region	Beijing	0.386	0.384	0.387	0.450	0.555	0.442	0.448	0.447	0.339	0.431
	Tianjin	0.262	0.291	0.284	0.203	0.310	0.320	0.330	0.333	0.260	0.330
	Hebei	0.200	0.246	0.215	0.291	0.437	0.252	0.269	0.264	0.456	0.259
	Shandong	0.291	0.350	0.326	0.287	0.464	0.340	0.352	0.355	0.341	0.367
	Jiangsu	0.431	0.454	0.448	0.304	0.532	0.452	0.458	0.455	0.414	0.466
	Shanghai	0.360	0.371	0.374	0.267	0.459	0.415	0.427	0.426	0.286	0.415
	Zhejiang	0.422	0.443	0.433	0.682	0.479	0.439	0.449	0.449	0.373	0.454
	Fujian	0.286	0.400	0.374	0.663	0.458	0.322	0.336	0.340	0.289	0.344
	Guangdong	0.461	0.488	0.483	0.648	0.557	0.503	0.510	0.499	0.600	0.508
	Hainan	0.251	0.258	0.247	0.578	0.448	0.258	0.258	0.270	0.268	0.273
	Liaoning	0.233	0.271	0.252	0.437	0.399	0.274	0.290	0.288	0.270	0.285
	Mean	0.326	0.360	0.348	0.437	0.463	0.365	0.375	0.375	0.354	0.376
Central region	Jiling	0.262	0.285	0.264	0.474	0.310	0.300	0.315	0.299	0.459	0.283
	Heilongjiang	0.319	0.349	0.329	0.526	0.359	0.352	0.363	0.351	0.473	0.338
	Shanxi	0.206	0.253	0.220	0.208	0.352	0.251	0.266	0.257	0.471	0.255
	Henan	0.193	0.241	0.215	0.263	0.328	0.226	0.247	0.254	0.303	0.254
	Hubei	0.217	0.271	0.240	0.394	0.365	0.269	0.282	0.276	0.441	0.275
	Hunan	0.208	0.256	0.227	0.525	0.342	0.255	0.271	0.274	0.380	0.269
	Anhui	0.174	0.223	0.197	0.323	0.339	0.214	0.233	0.235	0.253	0.227
	Jiangxi	0.248	0.287	0.261	0.618	0.500	0.265	0.283	0.291	0.285	0.290
	Mean	0.228	0.271	0.244	0.416	0.362	0.267	0.282	0.280	0.383	0.274
Western region	Guangxi	0.215	0.260	0.232	0.581	0.304	0.244	0.260	0.268	0.355	0.265
	Chongqing	0.227	0.262	0.237	0.426	0.416	0.269	0.283	0.284	0.463	0.275
	Sichuan	0.261	0.293	0.269	0.436	0.380	0.278	0.293	0.299	0.302	0.296
	Guizhou	0.185	0.254	0.216	0.385	0.203	0.238	0.251	0.260	0.367	0.256
	Yunnan	0.311	0.340	0.320	0.565	0.397	0.335	0.347	0.348	0.455	0.337
	Shaanxi	0.277	0.293	0.272	0.459	0.384	0.284	0.301	0.298	0.431	0.296
	Gansu	0.146	0.193	0.166	0.147	0.159	0.212	0.218	0.199	0.491	0.188
	Qinghai	0.413	0.379	0.370	0.231	0.336	0.380	0.379	0.394	0.579	0.395
	NeiMonggol	0.348	0.351	0.343	0.312	0.359	0.379	0.391	0.376	0.555	0.357
	Ningxia	0.216	0.255	0.227	0.178	0.376	0.277	0.288	0.277	0.569	0.265
	Xinjiang	0.200	0.242	0.212	0.140	0.326	0.227	0.247	0.252	0.325	0.247
	Mean	0.254	0.284	0.260	0.351	0.331	0.284	0.296	0.296	0.445	0.289
National mean		0.274	0.308	0.288	0.400	0.388	0.309	0.321	0.321	0.395	0.317

**Table 3 ijerph-20-01707-t003:** Comparison of green development in China from 2010 to 2014 and 2015 to 2019.

Region	2010	2011	2012	2013	2014	Growth Rate	2015	2016	2017	2018	2019	Growth Rate
	The First Stage						The Second Stage					
Eastern region	0.326	0.360	0.348	0.437	0.463	42%	0.365	0.375	0.375	0.354	0.376	2.9%
Central region	0.228	0.271	0.244	0.416	0.362	59%	0.267	0.282	0.280	0.383	0.274	2.6%
Western region	0.254	0.284	0.260	0.351	0.331	30%	0.284	0.296	0.296	0.445	0.289	1.7%
Nationwide	0.274	0.308	0.288	0.400	0.388	42%	0.309	0.321	0.321	0.395	0.317	2.5%

## Data Availability

Not applicable.

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
