# Peer review of "How to Evaluate the Level of Green Development Based on Entropy Weight TOPSIS: Evidence from China"

_ijerph, 2023, doi:10.3390/ijerph20031707_

Round 1

Reviewer 1 Report

This article discusses how to evaluate the level of green development, using the Entropy Weight Topsis method, which has some practical significance but there are already relevant studies and this article is not innovative enough, the research method is too single, the data is not rich and weak, the graphical analysis is too superficial, the scientificity of the conclusions needs to be improved, in addition to the article has detail errors. Problems exist as follows.

1. The author's keywords include the new development concept. It should be taken into account that the reader may not be familiar with the essence of China's new development concept policy. It is possible to explain in the paper what the implemented concept is based on, what theoretical background it has and in which planning and strategy documents it is reflected. Such an explanation would help to assess the extent to which the level of green development is developed in the various provisions.

2. I suggest adding what the universal significance of the research is.

3. The method used, with the serial numbers in its formula neatly arranged.

4. No clear assignment of index layers to criteria layer in lines in Table 1. Add principles for the selection of indicators and further explanation of the selection of indicators.

5. There are few illustrations and tables in this paper, and the data support is weak. The content of the charts could be enriched, while the analysis of the content is too superficial, and specific analysis of policy implementation and its effects could be added to increase the convincingness of the chart explanation.

6. The weakness of this paper is that it does not compare the Chinese approach with the planning and strategic practices implemented in other countries and regions. Compared with the experiences of other countries, the findings of the study would have made it possible to show which policies adopted for green development are sufficiently general to provide an opportunity to change the approach and which may not be feasible, taking into account differences in conditions.

7. The recommendations made in the concluding section (5) could be more specific.

Reviewer 2 Report

The manuscript titled: How to Evaluate the Level of Green Development Based on Entropy Weight TOPSIS: Evidence from China.

The aims and scope of this paper match those of the International Journal of Environmental Research and Public Health. The manuscript is interesting and well-organized. Based on my opinion, it needs some improvements to be considered for publication in IJERPH. I would suggest some changes that, in my opinion, would improve the paper:

. In lines 174-177, the authors state that the entropy weight TOPSIS is an improvement of the TOPSIS method. I would disagree with the authors on this claim. This is not about improving the TOPSIS method but about a classic hybrid model based on the Entropy method and the TOPSIS method.

. The flowchart of the methodology is missing. Explain the flowchart additionally through the text. It is not enough to explain only the methods you used, but how they are related. A flowchart should give readers a clear picture of how you arrived at your results.

. Divide equation 3 into two parts. Check the first part of equation 3.

. In equations 6 and 7, "ij" which is next to x`, should be in the subscript.

. Check equation 7.

. The CV index is not the clearest explanation. What does the "C" stand for in equatation 9? I indirectly understood that the CV index is calculated for each region and for each year (from figure 1). This should be written in section 3.2. Readers should not understand this indirectly but should make it clear to them from your explanation of the methodology.

. The presentation of the results in Section 4.1 is very modest.

. According to how they defined the model, i.e. applied the entropy method for calculating the weight of the index, it is concluded that the weight of the index is different for each province and for each year. In my opinion, this approach does not give good results, that is, it gives results that cannot be compared. Therefore, the author should write a clear explanation of why they used the Entropy method for the calculation weight of the index. (I think it would have been much better for this kind of research if the authors had used some subjective method for defining the weight of the index, such as AHP, DIBR, BWM, FUCOM, ect.)

. Sensitivity analysis is missing. I think that for this case study, an analysis by changing the weight of the index would be very useful. This should be shown, compared, and discussed. If good results were obtained through the sensitivity analysis, perhaps the application of the Entropy method could be justified. It is clear that this is a big work, but it is necessary to implement this procedure and demonstrate the quality of the results obtained.

. Papers of this type should include comparing the results with other studies (or other methods).

Round 2

Reviewer 1 Report

Strengthen Introduction and Literature Review.

Add "data availability statement".

Reviewer 2 Report

All the reviewers' comments have been addressed carefully and sufficiently. The revisions are rational from my point of view. I think the current version of the paper can be accepted.